# Deriving Motor States and Mobility Metrics from Gamified Augmented Reality Rehabilitation Exercises in People with Parkinson’s Disease

**DOI:** 10.3390/s25237172

**Published:** 2025-11-24

**Authors:** Pieter F. van Doorn, Edward Nyman, Koen Wishaupt, Marjolein M. van der Krogt, Melvyn Roerdink

**Affiliations:** 1Department of Human Movement Sciences, Faculty of Behavioural and Movement Sciences, Vrije Universiteit Amsterdam, Amsterdam Movement Sciences, 1081 BT Amsterdam, The Netherlands; 2Department of Nutrition and Movement Sciences, NUTRIM Institute of Nutrition and Translational Research in Metabolism, Faculty of Health, Medicine and Life Sciences, Maastricht University, 6211 LK Maastricht, The Netherlands; 3Department of Nutrition and Movement Sciences, MHeNs Institute of Mental Health and Neurosciences, Faculty of Health, Medicine and Life Sciences, Maastricht University, 6211 LK Maastricht, The Netherlands; 4Department of Data and AI, Strolll Ltd., Stafford ST16 2LP, UK; 5Department of Rehabilitation Medicine, Amsterdam UMC Location Vrije Universiteit Amsterdam, 1081 HV Amsterdam, The Netherlands; 6Rehabilitation & Development, Amsterdam Movement Sciences, 1081 HV Amsterdam, The Netherlands; 7Department of Science, Strolll Ltd., Stafford ST16 2LP, UK

**Keywords:** augmented reality, Parkinson’s disease, concurrent validity, mobility, gait, turning, transfers, motor states, gamified exercises

## Abstract

People with Parkinson’s disease (PD) experience mobility impairments that impact daily functioning, yet conventional clinical assessments provide limited insight into real-world mobility. This study evaluated motor-state classification and the concurrent validity of mobility metrics derived from augmented-reality (AR) glasses against a markerless motion capture system (Theia3D) during gamified AR exercises. Fifteen participants with PD completed five gamified AR exercises measured with both systems. Motor-state segments included straight walking, turning, squatting, and sit-to-stand/stand-to-sit transfers, from which the following mobility metrics were derived: step length, gait speed, cadence, transfer and squat durations, squat depth, turn duration, and peak turn angular velocity. We found excellent between-systems consistency for head position (X, Y, Z) and yaw-angle time series (ICC_(c,1)_ > 0.932). The AR-based motor-state classification showed high accuracy, with F1-scores of 0.947–1.000. Absolute agreement with Theia3D was excellent for all mobility metrics (ICC_(A,1)_ > 0.904), except for cadence during straight walking and peak angular velocity during turns, which were good and moderate (ICC_(A,1)_ = 0.890, ICC_(A,1)_ = 0.850, respectively). These results indicate that motor states and associated mobility metrics can be accurately derived during gamified AR exercises, verified in a controlled laboratory environment in people with mild to moderate PD, a necessary first step towards unobtrusive derivation of mobility metrics during in-clinic and at-home AR neurorehabilitation exercise programs.

## 1. Introduction

People with Parkinson’s disease (PD) experience motor symptoms that significantly impact daily activities and quality of life [1]. Monitoring mobility provides crucial insights into disease progression and aids clinicians in devising effective treatment plans. Clinical mobility tests like the Timed Up and Go (TUG), Five Times Sit-to-Stand (FTSTS), and 10 m walk test (10 MWT) are commonly used to assess mobility in people with PD [2]. The primary outcome of these tests is the completion time, which correlates with key mobility indicators, including muscle strength, gait speed, balance, and fall risk [3,4,5]. However, completion time itself does not provide specific information about underlying mobility characteristics, such as gait performance, turning ability, or squat performance.

Instrumented tests have, therefore, been developed to provide more detailed metrics such as gait speed, step length, turn duration, and durations of sit-to-stand and stand-to-sit transfers, allowing clinicians to monitor specific aspects of mobility more precisely [6]. Yet, these tests typically capture isolated snapshots of mobility and may introduce biases like the white-coat effect. Ideally, mobility should also be assessed in settings that reflect real-world performance rather than only capacity under controlled conditions, which has limited ecological value [7]. This distinction between unsupervised performance and supervised capacity could be relevant for understanding the true impact of mobility difficulties in daily life [8].

Augmented-reality (AR) neurorehabilitation platforms can engage individuals with PD in gait-and-balance-enhancing exercises in a gamified manner. An advantage of using AR technology is the continuous recording of 3D positional and orientation data of the AR glasses using visual simultaneous localization and mapping (vSLAM) algorithms [9], which directly serve as a 6D motion capture device of the head. Previous studies have validated the use of this data to derive various mobility metrics during standard clinical tests (e.g., FTSTS, TUG, and 10 m walk test), demonstrating good-to-excellent agreement with reference systems in people with PD [10,11]. Furthermore, several studies have already examined the usability of AR glasses in people with PD and found that they are generally well-tolerated and safe to use, supporting their feasibility for less burdensome mobility assessments from a form-factor, safety, and wearability perspective [12,13].

Extending the use of AR glasses to capture 6D head movements during gamified exercises and translating these movements into specific mobility metrics offers several advantages. First, patients can receive real-time feedback on their mobility performance during exercises, an aspect that patients themselves reported as valuable during exercises, and that offers potential for self-management [12]. Second, clinicians gain access to more detailed metrics without the need for resource-intensive and time-consuming mobility tests in the clinic. Third, this approach could potentially enable remote real-time derivation of mobility metrics over time, capturing continuous, objective data on motor performance during daily exercises. This could address the gap highlighted by Negi et al. (2025), who reported that infrequent clinic visits and reliance on subjective measurements limit effective clinical management and that by remotely tracking progression over time, subtle changes in motor function can be detected outside the clinic, allowing interventions to be optimized in a timely and personalized manner [14]. This potential could significantly reduce the growing burden on the healthcare system by decreasing the need for in-person visits and prioritizing assessments and interventions for those most in need [15].

A critical step in deriving mobility metrics is accurate detection of motor states, such as walking, turning, squatting, and transfers. Identifying these states enhances the extraction of specific parameters within those motor states, such as gait speed, step length, cadence, turning duration, peak turn velocity, squat duration, squat depth, and sit-to-stand and stand-to-sit durations. The primary aim of this work is to evaluate the concurrent validity of AR-derived motor states and associated mobility metrics by comparing them with a markerless motion capture system (Theia3D). Markerless motion capture systems have generally shown acceptable validity for gait assessment, and Theia3D in particular produces spatiotemporal gait parameters comparable to marker-based systems [16,17]. A recent study by Seuthe et al. (2025) reported good-to-excellent agreement between Theia3D-derived and marker-based-derived gait measures in people with PD [18]. Secondly, this study introduces a reproducible and publicly available technical framework for automated motor-state segmentation and subsequent mobility metric derivation from AR data.

## 2. Materials and Methods

### 2.1. Subjects

A convenience sample of 15 subjects diagnosed with Parkinson’s disease, who previously participated in the study by Harderman et al. (2024) (N = 4) [19] or the study by Hoogendoorn et al. (2025) (N = 11) [13] and were, therefore, familiar with gamified AR exercises, participated in this study (Table 1). Participants did not have any other neurological or orthopedic disorders that would significantly impact their mobility. All participants had sufficient cognitive function and could understand and follow the instructions provided by the researchers. None reported experiencing hallucinations or having any visual or hearing impairment that could affect gamified AR exercises.

### 2.2. Experimental Setup and Procedures

Participants were invited for one visit to the gait lab at the Amsterdam University Medical Center (Amsterdam UMC), location VUmc, and were instructed to take their Parkinson’s medication/dopaminergic medication 1 h prior to the visit. Participants completed 5 gamified AR exercises in a fixed order: (i.e., *Smash*, *Mole Patrolll*, *Puzzle Walk*, *Basketball*, and *Wobbly Waiter*) for 3 min each using Strolll’s AR neurorehabilitation platform (Figure 1 and Appendix A). This platform is specifically designed for people with Parkinson’s disease to improve gait and balance through gamified, personalized, and accessible exercises that can be performed both in the clinic and at home. Participants could take a break between games whenever needed. For each 3 min game, a portion of the gameplay was recorded with the Theia3D system, limited to 30–60 s due to processing issues with Theia3D when recording longer time periods.

### 2.3. Data Acquisition

The AR glasses used were Magic Leap 2 (Magic Leap, Inc., Plantation, FL, USA), which captured the user’s 3D position and orientation with regard to their surroundings at a sample rate of 60 Hz. Video data for Theia3D were collected for each game for 30 s with a sample rate of 100 Hz, or 1 min with a sample rate of 50 Hz. The video data were collected using 7 Blackfly S USB3 cameras (Teledyne FLIR LLC, Wilsonville, OR, USA; resolution: 1920  ×  1200 pixels, 2.3 megapixels; focal length: 6.5–15.5 mm), which were positioned around a 10 m walkway at a distance of 2 to 3 m (Figure 2). Calibration for the camera system was performed before each measurement day to establish a consistent global reference frame.

### 2.4. Data Analysis

For the markerless processing, video data were transcoded using the h.264 codec within Vicon Nexus software (version 2.16, Vicon, Oxford, UK). The transcoded video files were then processed using Theia3D software (v2023.1.0.3160, Theia Markerless Inc., Kingston, ON, Canada) with default settings, including a 6 Hz low-pass GVCSPL filter. The Theia3D default inverse kinematic model was applied. The output consisted of 4  ×  4 transformation matrices for each body segment. For our analysis, only the virtual markers of the segments of the head, pelvis, heels, and toes were used [21]. No additional Theia3D internal processing steps, such as event detection or metric computation, were used in the motor-state detection or in the derivation of mobility metrics. The time series from both the Magic Leap 2 and Theia3D were resampled to a constant rate of 50 Hz using linear interpolation. Time series were smoothed using a Savitzky–Golay filter with a second-order polynomial and a window size of 61 frames. This filtering method was chosen because it effectively smooths data while preserving the original shape and features of the signal—such as peaks, troughs, and slopes—used in defining start and end indices of a motor [22]. Missing values in the Theia3D time series were interpolated using cubic spline interpolation. One trial (*Mole Patrolll*) from a single participant (P8) was excluded for further analysis because more than 10% of the Theia3D data were missing due to the participant moving outside the camera capture range. Temporal alignment between systems was achieved by calculating the time lag of the maximal cross-correlation of the vertical position of the head from both systems. The consistency agreement between 3D position data of the AR glasses and Theia3D head and between the yaw angle of the AR glasses and Theia3D pelvis was determined separately for each game. Theia3D pelvis yaw was used as a reference for the turning motor state because it provides a more stable and anatomically relevant measure of whole-body rotation, whereas head yaw could be more susceptible to non-turning-related head motions, such as scanning. These additional head movements could inflate angular velocity estimates and introduce noise, potentially reducing the accuracy of turning performance. For all other motor states, the Theia3D head data was used as the reference.

### 2.5. Motor-State Segment Classification and Derived Mobility Metrics

For each motor-state segment detected in the AR data, the temporal overlap with segments detected in Theia3D was evaluated. Overlap was defined as the proportion of AR frames that coincided with frames from a corresponding Theia3D segment. For each AR segment, the Theia3D segment with the highest overlap ratio was identified as the best match. A match was considered valid if the best overlap ratio was ≥50% of the AR segment. True positives were defined as AR segments with a corresponding Theia3D match, false positives as AR segments without a matching Theia3D segment, and false negatives as Theia3D segments without a matching AR segment.

Mobility metrics were determined only for true positive segments, and only the fully overlapping frames from both systems were used to calculate these parameters. The following sections outline in detail the methods used for identifying these motor-state-specific parameters. A general overview flowchart of motor-state classification, segment matching, and mobility metric derivation, as well as pseudocode for all algorithms, is available on our GitHub repository (https://github.com/pvand/Motorstate_detection_and_metrics).

#### 2.5.1. Straight-Walking Segments and Gait Metrics

Straight-walking segments for both systems in all games were identified by applying a custom algorithm to the horizontal position (x, y) and yaw angle time series of the head. The horizontal trajectory was first used to calculate arc length and frame-to-frame speed. To ensure a smooth and consistent heading direction, the trajectory was interpolated along uniform 0.01 m steps. The heading angle, defined as the instantaneous orientation of the horizontal trajectory in the plane, was derived from the tangent (first derivative) of the interpolated trajectory using the arctangent of the forward and lateral components, and subsequently mapped back to the original resolution. This processing was necessary to reduce noise in the heading angle.

Candidate straight-walking segments were identified based on the following criteria: (1) the total segment length exceeded 2 m; (2) the range of heading angles remained below 45 degrees; (3) the range of head yaw angles was below 45 degrees; and (4) all instantaneous speeds within the segment exceeded a minimum speed threshold of 0.5 m/s (Figure 3a).

Maximum gait speed was derived from AR and Theia3D straight-walking segment data as the highest instantaneous speed, computed from frame-to-frame 2D (x, y) head displacement over time. To reduce high-frequency fluctuations in the frame-by-frame speeds, a moving average filter with a window size of 0.5 s was applied. The maximum gait speed was then defined as the peak value of the filtered speed, calculated separately for each straight-walking segment for both the AR and Theia3D systems.

To estimate mean step length from Theia3D, heel-strike (HS) events were first detected using the minimal distance between heel and toe (MDHT) approach [23]. This involved calculating the vertical difference between heel and toe positions. Candidate HS events were detected by identifying the local maxima in the MDHT signal using Matlab’s findpeaks with a prominence threshold of 0.01 and a minimum peak distance of 0.5 s (Figure 3c). For each detected peak, the closest preceding minimum within a defined search range (0.05 to 0.5 s) was identified. If this minimum was at least 0.04 m below the peak and had an absolute value lower than 0.05 m, it was retained as a valid heel strike (HS). To avoid duplicate detections, consecutive minima at the same location were ignored. Step length was then calculated based on the Euclidean horizontal (x, y) Theia3D head segment displacement between consecutive HS events (Figure 3b).

The mean step length for AR data was calculated using the vertical position time series. The signal was linearly detrended, and a Butterworth bandpass filter with cutoffs at 1.5 and 3.5 Hz (based on the expected cadence) was applied to detect indices at or around the double support (DS) phase of the gait cycle. Local minima in the vertical position of the head, representing DS [24], were found using Matlab’s findpeaks with a minimum peak prominence of 0.001 and a minimum peak distance of 13 samples. Finally, the step length was calculated using the Euclidean horizontal (x, y).

Cadence (steps per minute) was estimated for both Theia3D and the AR glasses based on the timing of detected gait events. For Theia3D, this was performed by calculating the median number of frames between consecutive heel-strike events and converting this value into steps per minute. For AR, the same approach was used based on the detected DS events.

#### 2.5.2. Turning Segments and Turn Metrics

For one game (*Smash*), turn metrics were determined. In this game, a 180-degree turn is required/performed (Figure 4a). The merged turn algorithm described by Shah et al. (2021) was used to estimate the start and end of a fixed 180-degree turn based on the Theia3D pelvis yaw and the AR glasses yaw data [25]. The pelvis yaw angle from Theia3D was specifically used, instead of the head yaw angle from Theia3D. This was performed to not arbitrarily inflate between-systems agreement, thereby conservatively evaluating the AR-system’s ability to distinguish full-body turns from head-only movements, which is not arbitrary. Since this algorithm of Shah and colleagues relies on thresholds based on yaw angular velocity, the derivative of the yaw angle was computed. Because the original method was developed for real-world turning detection—typically involving slower, less abrupt turns—the thresholds were empirically optimized to better reflect the faster, more pivot-like turning observed in the game context (Table 2) (Figure 4b). Different thresholds were used for Theia3D pelvis yaw data and AR head yaw data to account for signal characteristics, with head yaw angles generally showing more variability and sharper angular changes than pelvis-based data (Figure 4c).

To determine the optimal AR thresholds, an iterative error-cost approach was applied. A range of threshold combinations was systematically tested, and for each combination, turn durations derived from AR were compared against Theia3D pelvis-based turn durations. The intraclass correlation coefficient (ICC_(A,1)_) was used as the optimization criterion, and the threshold set that produced the highest ICC value was selected as the final parameterization for AR (Table 2). The mean of the turn durations and the peak turn velocity per participant were extracted for further analysis (Figure 4c).

#### 2.5.3. Squat Segments and Squat Metricss

In one game (*Basketball*), squatting is an integral part of gameplay. To identify squats, the vertical head position from both the Theia3D and AR system was analyzed. Local minima were identified using Matlab’s findpeaks on the inverted vertical signal. These minima were clustered using k-means (k = 2, iteration = 10) to distinguish deeper valleys (i.e., squats) from shallower ones (e.g., bending forward or minor dips) (Figure 5a). Valleys that occurred close together (<8 s) were grouped to account for consecutive squats performed in quick succession. Within each group, the vertical velocity signal was used to identify the start and end of individual squats. By searching backwards from each detected valley, the start of the squat was marked at the last time point where the downward velocity reached the zero-crossing. Searching forwards from the valley, the end of the squat was marked at the first time point where upward velocity reached the zero-crossing (Figure 5b).

The duration of each squat was then calculated. Squat depth was defined as the difference between the baseline vertical head position (average of the start and end positions) and the minimum vertical head position within the squat. Finally, the mean squat depth and mean squat duration per participant were used for further analyses.

#### 2.5.4. Transfer Detection and Transfer Metrics

In one game (*Wobbly Waiter*), sit-to-stand and stand-to-sit transfers are key gameplay motor states. To identify transfers, standing segments were first detected using k-means clustering (k = 2, iteration = 10) on the vertical head position data from both the Theia3D and AR systems. The cluster with the highest centroid (i.e., highest vertical position) was classified as standing (Figure 6a). For each standing segment, the onset and offset of the sit-to-stand and stand-to-sit transfers were detected in the vertical speed signal using a 0.001 m/s threshold (Figure 6b). Transfer durations were averaged per participant and taken for further analysis.

### 2.6. Statistical Analysis

Concurrent validity between AR and Theia3D time series was evaluated with the intraclass correlation coefficient for consistency ICC_(C,1)_, and the root mean square error (RMSE). Before calculating these metrics, the data were preprocessed to account for differences in alignment due to different coordinate systems. Both AR and Theia3D signals were detrended to remove constant offsets.

For each detected motor state, performance parameters, i.e., precision, recall, F1-score, and mean overlap, were calculated.

Concurrent validity in mobility metrics derived from AR and Theia3D data was evaluated with the intraclass correlation coefficient for absolute agreement ICC_(A,1)_ [26]. ICC values above 0.50, 0.75, and 0.90 were considered as moderate, good, and excellent agreement, respectively [27]. ICC values for absolute agreement were complemented with bias and limits of agreement measures, as calculated using a Bland–Altman analysis [28]. In addition, the standard error of measurement (SEM) was calculated from the SD and ICC_(A,1)_ to provide an estimate of absolute measurement precision. Biases were evaluated using paired-sample t-tests under the verified assumption of normality using the Shapiro–Wilk test. If data were not normally distributed, the Wilcoxon signed-rank test was used to evaluate biases. Statistical analyses were run in Matlab version 2023b, with significance set at 0.05.

## 3. Results

### 3.1. Consistency Agreement in Time Series Between Systems

In Figure 7, the time series for both systems is shown for each axis and for each game.

At a group level, the consistency agreement between Theia3D head and AR planar X and Y positions was excellent for all games (ICC_(C,1)_ > 0.932; RMSE 0.005–0.080 m). The consistency agreement between Theia3D head and AR vertical Z position was good to excellent (ICC_(C,1)_ = 0.736–0.998; RMSE 0.011–0.027 m). The consistency agreement between Theia3D pelvis yaw angle and AR yaw angle was excellent (ICC_(C,1)_ > 0.957; RMSE 12.0–21.6°), except for *Basketball* (ICC_(C,1)_ = 0.335; RMSE 5.44°) (Table 3).

### 3.2. Motor-State Segment Classification 

For straight walking, 130 segments were detected in Theia3D and 134 in AR, with 125 true positives, 5 false positives, 9 false negatives, a precision of 0.962, a recall of 0.933, an F1-score of 0.947, and a mean overlap of 97.1% ± 0.3%. Pivot turning had 114 segments detected in both systems, with 113 true positives, 1 false positive, 1 false negative, a precision and recall of 0.991, an F1-score of 0.991, and a mean overlap of 78.3% ± 0.9%. Squatting was detected for 134 segments with perfect precision and recall (1.000) and a mean overlap of 97.2% ± 0.6%. Sit-to-stand and stand-to-sit movements were also detected for 39 and 40 segments, respectively, with perfect precision and recall (1.000), with mean overlaps of 90.8% ± 0.4% and 93.1% ± 0.8%, respectively (Table 4).

### 3.3. Absolute Agreement in Mobility Metrics

The absolute agreement between Theia3D head and AR mobility metrics was moderate to excellent. For gait metrics, mean step length and maximum gait speed showed excellent agreements (ICC_(A,1)_ = 0.938 and ICC_(A,1)_ = 0.999, respectively) without systematic bias, and cadence showed good agreement (ICC_(A,1)_ = 0.850) also without systematic bias. For turn metrics, turn durations showed excellent agreement (ICC_(A,1)_ = 0.904) without systematic biases, and they were moderate for peak angular velocity (ICC_(A,1)_ = 0.477) with a statistically significant bias (~18% of the mean) (Table 5). For transfer metrics, both sit-to-stand and stand-to-sit mean durations showed excellent agreement (ICC_(A,1)_ > 0.978) without systematic bias. For squat metrics, squat duration showed excellent agreement (ICC_(A,1)_ = 0.995) without systematic bias, while squat depth also showed excellent agreement (ICC_(A,1)_ = 0.969) but with statistically significant bias (~9% of the mean). All metrics had comparable SEMs within systems (Table 5).

## 4. Discussion

The aim of this study was to evaluate the concurrent validity of AR time series, classified motor states, and associated mobility metrics derived from AR glasses data against a markerless motion capture system (Theia3D) during gamified AR exercises in people with Parkinson’s disease. Overall, we found excellent between-systems consistency for head position (X, Y, Z) and yaw angle time series; high accuracy for AR classified motor states, straight walking, turning, squatting, and transfers; and excellent absolute agreement for all mobility metrics, except for cadence and peak angular velocity. These findings are discussed below with reference to the state of the art, the strengths and limitations, and the implications for future work.

### 4.1. Interpretation and Comparison with Other Literature

#### 4.1.1. Consistency Agreement in Time Series

The consistency in planar X and Y positions between AR and Theia3D was excellent for all games (ICC_(C,1)_ > 0.932; RMSE ≤ 0.08 m; Table 3), confirming the accuracy of the AR glasses’ tracking capabilities using vSLAM [8]. In the vertical Z position, the consistency was excellent for games with substantial vertical head displacement. For games with limited vertical motion (i.e., *Smash* and *Mole Patrolll*; ICC_(C,1)_ = 0.736 and 0.780, respectively), RMSE values were also slightly higher (0.011–0.016 m), reflecting the smaller amplitudes and the greater relative influence of noise on between-systems agreement (cf. Figure 7). When comparing the pelvis yaw angle from Theia3D with AR yaw angle, we found excellent consistency across all games except *Basketball*, which showed low agreement (ICC_(C,1)_ = 0.335; RMSE 5.438°) due to the absence of turning—the participants remained stationary while squatting and shooting, orienting their head to the rack of balls and the hoop, which were in the same line of sight. Overall, the strong agreement between pelvis and AR yaw highlights the potential of AR-derived head yaw as a surrogate for estimating turn parameters, offering a practical alternative to the commonly used lower-back yaw angle [29].

#### 4.1.2. Motor-State Segment Classification

Classified motor states (straight walking, turning, squatting, sit-to-stand, and stand-to-sit) with AR glasses showed excellent agreement compared to Theia3D, underscoring its ability to differentiate distinct motor behaviors in people with Parkinson’s disease. That said, some false positives (N = 5) and false negatives (N = 9) were identified for straight walking. These misclassifications are largely attributed to edge cases with respect to the threshold criteria applied for straight-walking segments: (1) total segment length > 2 m, (2) heading angle range < 45°, (3) head yaw range < 45°, and (4) all instantaneous speeds > 0.5 m/s. When edge cases within 5% of these thresholds were included, all false positives and eight of the nine false negatives were correctly reclassified, resulting in a precision of 0.993, a recall of 1.000, and an F1-score of 0.966. To put this in perspective, this would correspond to only 0.6% of motor-state segments being falsely identified. Our results align with the state of the art for gait classification in a real setting using a head-worn inertial sensor and feature-based machine learning [30]. Likewise, for sit-to-stand and stand-to-sit transfer segments, Zijlstra et al. (2012) also reported high accuracy of body-fixed sensor-based transfer durations against counterparts derived from force plates in patients with Parkinson’s disease [31]. Also, high accuracy in turning segments in people with Parkinson’s disease in a real-world setting has been shown using a single lower-back-worn inertial sensor [32,33,34]. Taken together, these studies and our new findings collectively underscore the potential of wearable sensors, including AR glasses, for classifying motor states in individuals with Parkinson’s disease.

#### 4.1.3. Absolute Agreement in Mobility Metrics

Excellent absolute agreement was found for mean step length and maximum gait speed, with no systematic bias, while cadence showed good agreement with a small, insignificant bias. However, the limits of agreement were relatively wide. This may be due to the different methods used between systems for cadence calculation. For Theia3D data, heel-strike events (via the MDHT algorithm) were used, whereas for AR data, cadence was estimated from minima in vertical head position, which occurs in the double-support phase [24]. Since these approaches rely on different events—heel strikes directly measure foot–ground contact, while head-based double support is an indirect and less sharply demarcated event influenced by head movement or postural sway—some variation between systems was to be expected. This likely explains the wider limits of agreement for cadence. Similar observations have been reported for AR-derived gait parameters compared with motion capture during the 10 m walk tests [11]. In addition, Seuthe et al. (2025) also found lower agreement for temporal gait parameters than for spatial gait parameters, highlighting the sensitivity of temporal measures to differences in event detection methods [18]. For transfer and squat metrics, excellent absolute agreement statistics were observed for sit-to-stand, stand-to-sit, and squat durations, without systematic biases except for squat depth (~9% of the mean). This bias is likely due to differences in the definition of the coordinate system origins: the AR glasses use a point located at the front, inside the glasses, whereas Theia3D defines the head origin around the midpoint between the ears [21]. Because the AR origin sits more anterior, pitch rotations during squats exaggerate vertical head displacement compared to Theia3D. In addition, small misplacements or tilts of the headset in the sagittal plane may have further amplified this effect. Consequently, vertical head displacement may be exaggerated in AR data because of its more anterior point of origin compared to Theia3D, and caution is warranted when interpreting vertical displacements solely based on AR glasses when head pitch rotations are involved, as discrepancies in origin placement of the AR glasses and in the center of rotation can substantially affect this accuracy. Similar findings for AR transfer metrics have been reported before for transfers involved in the Five Times Sit-to-Stand test [10].

For turn metrics, excellent absolute agreement was found for turn duration without systematic bias, indicating that the duration of a 180-degree pivot turn, even though it relied on head yaw rather than the more commonly measured pelvis or trunk yaws [35]. It should be noted, however, that defining the exact start and end of a turn is somewhat nebulous, as turning is characterized by a rather slow continuous transition rather than a sharply demarcated event. This result aligns with the findings of Rehman et al. (2020), who investigated the influence of sensor location on turning characteristics in the classification of Parkinson’s disease. In their study, inertial sensors were attached to both the head and lower back of 37 individuals with Parkinson’s disease. They demonstrated that turn metrics derived from head-mounted inertial sensors were comparable to counterparts obtained from the lower back, especially when combining spatiotemporal and signal-based features [34]. These results reinforce the notion that the head can serve as a reliable proxy for assessing the durations of 180-degree pivot turns, provided that the analytical approach accounts for the distinct movement patterns and potential noise associated with head motion (e.g., nodding, dyskinesia, and scanning). However, peak angular velocity showed only moderate agreement between Theia pelvis and AR head jaw, with a significant bias of ~18% of the mean, as AR-derived values were consistently higher than those from Theia3D. This discrepancy is likely attributable to non-turn-related head movements, such as looking around or scanning the environment, which inflate the measured angular velocity. That is, we find an excellent agreement for peak angular velocity (ICC_(A1)_ = 0.937), with only a small significant bias of 3.5% of the mean, when using the Theia3D head instead of the pelvis segment for peak angular velocity. These findings indicate a major limitation of using head-worn sensors for measuring whole-body turns: while durations of turns can be reliably captured, measures that depend on instantaneous or peak velocities are vulnerable to head movements unrelated to the whole-body turn. Consequently, peak angular velocity derived from head-worn AR sensors should be interpreted cautiously.

### 4.2. Strengths and Limitations

A major strength of this study lies in its innovative use of AR glasses as a continuous and unobtrusive tool to capture mobility metrics during gamified AR exercises. By comparing AR-derived mobility parameters with those obtained from a markerless motion capture system (Theia3D), we provide robust evidence supporting the concurrent validity of AR technology for deriving mobility metrics for various motor states in people with Parkinson’s disease. The inclusion of multiple gamified AR exercises promoting diverse motor states (e.g., straight walking, turning, squatting, and transfers) enhances the generalizability of our findings to different functional tasks relevant in daily life.

However, some limitations must be acknowledged. The most important limitation is the relatively small and homogenous sample consisting of 15 persons with PD (modified Hoehn and Yahr stages 1–2), which may limit the generalizability to the broader PD population, especially those at more advanced disease stages. However, our sample is representative of the intended target population for gamified AR-based exercises, as individuals with mild-to-moderate PD are generally able to perform these exercises independently and more safely [13,19]. Second, the study lacked a definitive ground truth for motor-state detection. Without labeled reference data, the validation of AR-derived outcomes relies on indirect comparisons rather than true state-by-state agreement. Establishing ground-truth labels through, for example, expert video annotation and applying supervised machine-learning methods could potentially provide a more robust basis for validating AR-derived motor states. Unfortunately, the current dataset was not large enough to support such supervised machine-learning approaches. Finally, while Theia3D serves as a well-validated markerless motion capture system for accurately estimating the position and orientation of body segments, the subsequent calculation of mobility metrics depends heavily on the specific algorithms and computational pipelines applied to this data. These algorithm pipelines involve steps such as signal filtering, segmentation of gait cycles, event detection (e.g., heel strikes), and parameter extraction. As a result, differences in parameter thresholds or processing assumptions may influence the derived mobility metrics, potentially affecting their accuracy and comparability with other measurement systems or across studies. We, therefore, explicitly reported all parameters, made our dataset publicly available, and provided general flow charts of the processing steps along with pseudocode of the algorithms used in this study.

### 4.3. Implications and Future Work

This study contributes to the growing body of evidence supporting the use of AR glasses as a promising tool for digital mobility assessment in people with PD. Beyond the feasibility of deriving valid mobility metrics from AR-based head tracking during standardized clinical tests [10,11], these findings underscore the potential of capturing mobility metrics during daily gamified AR exercises, which could facilitate more frequent, low-burden assessments of mobility.

One prospect of our findings lies in the integration of remote mobility monitoring within home-based AR-rehabilitation programs prescribed by clinicians. Remote monitoring of mobility outcomes could enable clinicians to track progress more frequently, enhancing personalized treatment and allowing for timely adjustments to interventions based on mobility metrics frequently captured in the real world. This approach aligns with the findings of Negi et al. (2025), who highlighted that infrequent clinic visits and reliance on subjective assessments create a critical gap in Parkinson’s disease management [14], and that continuous, objective digital monitoring can fill this gap by capturing subtle changes in motor functioning over time, allowing for timely and personalized intervention [13]. Emerging evidence further indicates that digital mobility outcomes (DMOs) derived from real-world assessments capture complementary aspects of functional mobility not observable during short, supervised clinical tests. Real-world DMOs, such as walking speed and step length, are often lower than laboratory measures, reflecting the challenges of everyday environments. Their moderate-to-weak correlations with clinical tests suggest that they provide distinct and valuable insights into functional mobility. This distinction between real-world performance and supervised capacity may, therefore, be valuable for understanding the true impact of mobility difficulties and for tailoring rehabilitation [8].

Another prospect is the possibility of presenting direct feedback on mobility performance during gamified AR exercises to the patients themselves. Providing such feedback in real time could increase user engagement and adherence. Feedback tailored to individual mobility characteristics—such as step length, walking speed, or turn, squat, and transfer durations—can help users monitor their own (progress in) performance. This type of feedback was also considered valuable by patients themselves when partaking in 6 weeks of gamified AR exercises at home [12]. Another study by Boege et al. (2024) emphasizes the potential of self-management systems, enabling patients to actively track and respond to their own functional performance, which enhances engagement, adherence, and empowerment in Parkinson’s disease care [36]. Additionally, real-time adjustments or in-game feedback based on ongoing movement data may support more effective training by identifying deviations from intended movement patterns, such as incomplete squats or irregular turning, and prompting corrective actions when needed.

Future research should explore the sensitivity and responsiveness of AR-derived mobility metrics for disease progression and treatment effects through longitudinal studies, for example, over a 6-week AR-rehabilitation program at home [13]. Examining changes in mobility metrics over time during a gamified AR exercises program—such as the changes in transfer durations, turning durations, or subtle declines in walking speed—could determine whether AR glasses can reliably capture clinically meaningful changes in mobility. Furthermore, establishing ground-truth labels through expert video annotation and applying supervised machine-learning methods would enable more accurate validation of AR-derived motor states and mobility metrics. Demonstrating both sensitivity to change and validity through such approaches would strengthen the potential of AR glasses for long-term, unobtrusive monitoring in people with Parkinson’s disease.

## 5. Conclusions

This study demonstrates that position and orientation data measured with AR glasses agree excellently with markerless motion capture data. From this data, specific motor states and associated mobility metrics can be accurately derived in people with mild-to-moderate Parkinson’s disease (H&Y stages 1–2) during multiple gamified AR exercises. These findings support the feasibility of the use of AR glasses as a tool for capturing detailed mobility metrics beyond traditional standardized clinical tests. This approach could potentially facilitate the derivation of mobility metrics in the clinic and remotely at home. Further research is warranted to evaluate the ecological validity, sensitivity, and responsiveness of these metrics to disease progression and treatments.

## Figures and Tables

**Figure 1 sensors-25-07172-f001:**
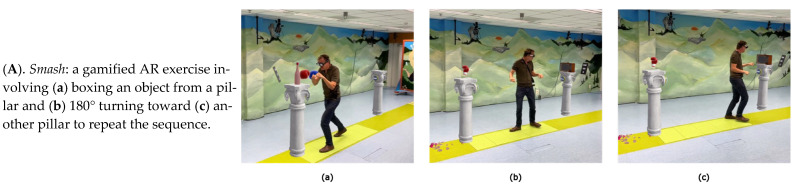
(**A**–**E**) Video stills from all used gamified AR exercises.

**Figure 2 sensors-25-07172-f002:**
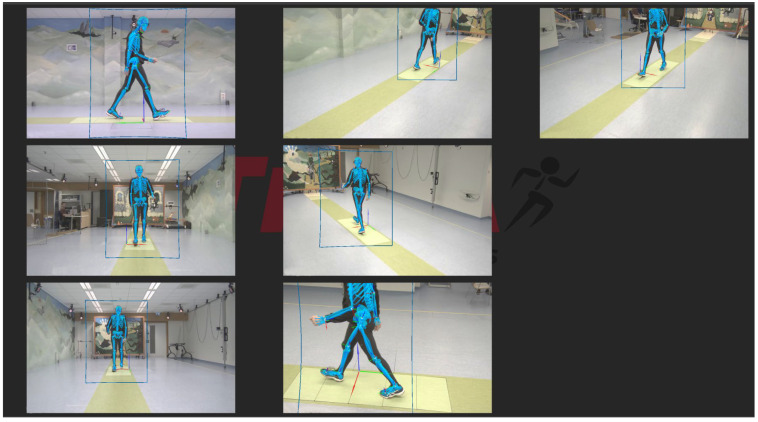
Experimental setup, with the output of the 7 video cameras and the Theia3D overlay.

**Figure 3 sensors-25-07172-f003:**
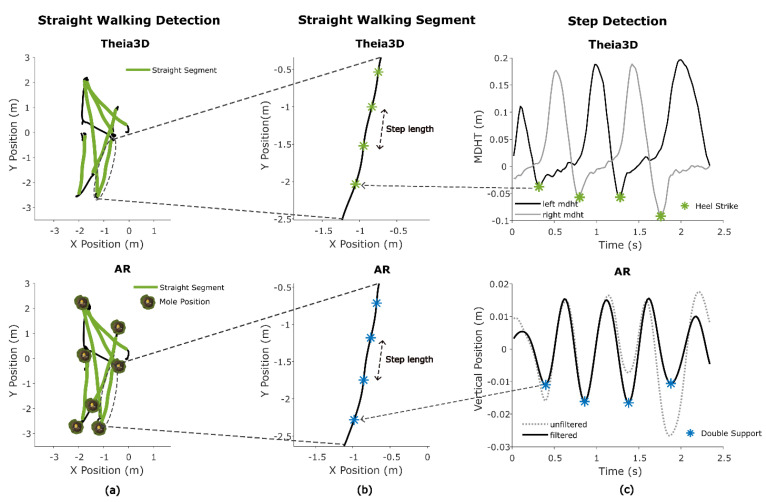
Straight-walking segments and gait metrics. (**a**): Detection of straight-walking segments within the 2D-trajectory using specific criteria. (**b**): Example of a straight-walking segment and the derivation of step length. (**c**): Heel-strike detection using the minimal distance between heel and toe (MDHT) approach, and double support detection using minima in the bandpass-filtered AR vertical position.

**Figure 4 sensors-25-07172-f004:**
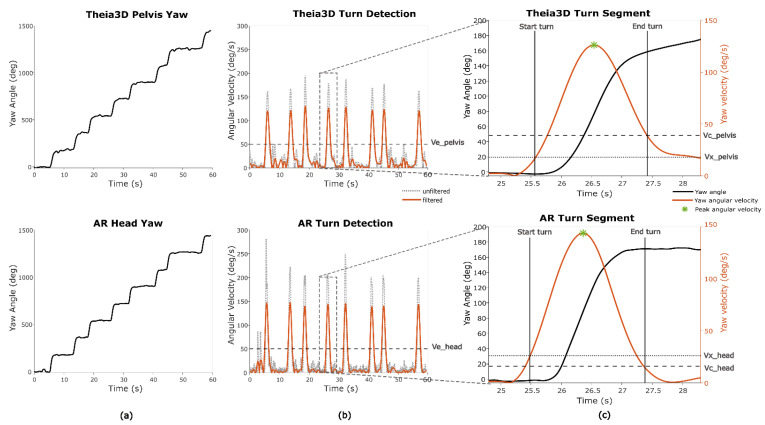
Turn segments and turn metrics during *Smash*. (**a**) Yaw angle of both systems. (**b**) Filtering of yaw angular velocity and detection of turns based on a shared turn threshold (Ve). (**c**) Detection of on- and offset of turns based on edge thresholds for pelvis and head (Vx and Vc) and peak angular velocity as the local maximum.

**Figure 5 sensors-25-07172-f005:**
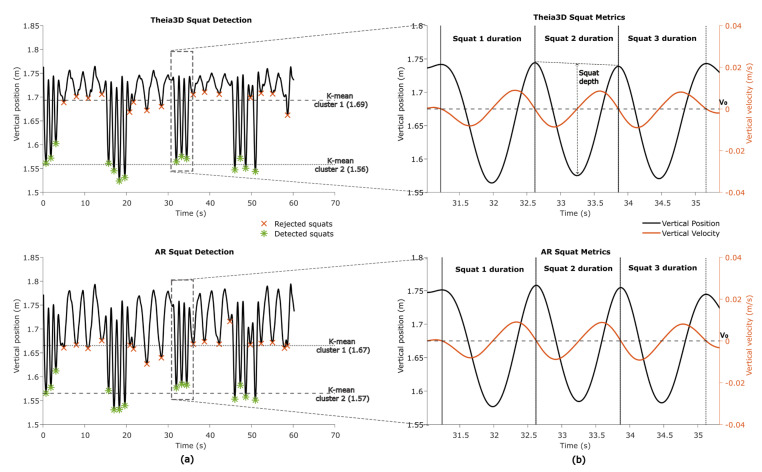
Squat segments and squat metrics. (**a**) Squat detection using k-means clustering of found minima. The cluster of minima with the lowest mean is identified as squats (cluster 2). (**b**) Squat duration using zero-crossing (V0) of the vertical velocity and squat depth derivation using vertical position.

**Figure 6 sensors-25-07172-f006:**
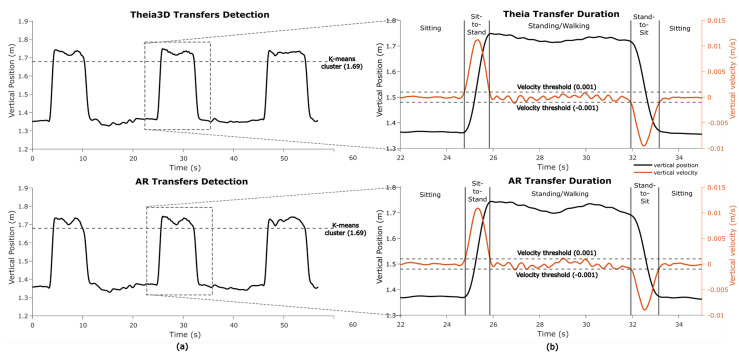
Transfer segments and transfer duration in *Wobbly Waiter*. (**a**) Detection of transfers using k-means clustering to distinguish between standing and sitting motor states in the vertical position. (**b**) Detection of transfer durations using thresholds in the vertical velocity.

**Figure 7 sensors-25-07172-f007:**
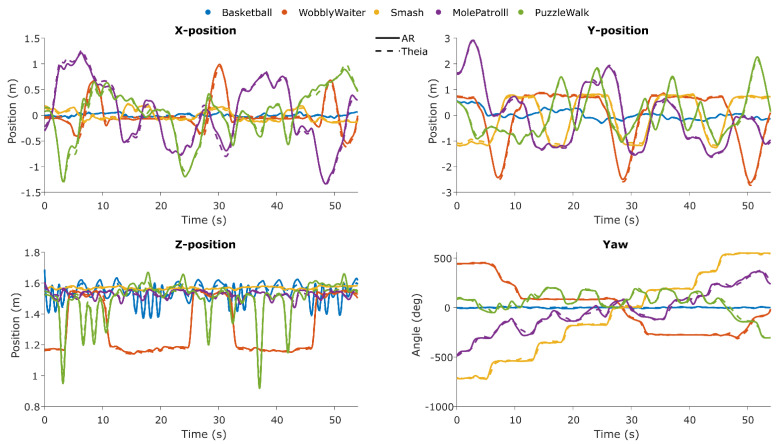
Time series of Theia3D and AR for each game and each used axis, showcasing consistency between systems and the level of variability of each axis per game.

**Table 1 sensors-25-07172-t001:** Participants’ characteristics.

Characteristics	Mean ± SD [Range] or No.
Age (years)	66.8 ± 6.5 [57–78]
Weight (kg)	85.4 ± 7.3 [75–102]
Height (cm)	180.3 ± 10.4 [164–196]
Sex, male/female	11/4
Time since diagnosis (years)	7.4 ± 5.6 [2–21]
Modified Hoehn and Yahr [20] stage, 1/2	4/11

**Table 2 sensors-25-07172-t002:** Thresholds and filter parameters for turn segments and turn metrics.

Description	Theia3D Pelvis Value	AR Head Value
Detection filter impulse response duration	1.5 s	1.5 s
Required depth for minima	20°/s	20°/s
Required velocity peak to detect turn (Ve)	50°/s	50°/s
Required velocity peak to detect edge (Vx)	18°/s	26°/s
Required velocity peak to detect edge (Vc)	39°/s	16°/s

**Table 3 sensors-25-07172-t003:** ICC consistency and RMSE of Theia3D head 3D position compared to AR 3D position, and Theia3D pelvis yaw angle compared to AR yaw angle per game at the group level.

Game	Value	X-Position ICC_(C,1)_ (95% CI)	Y-Position ICC_(C,1)_ (95% CI)	Z-Position ICC_(C,1)_ (95% CI)	Yaw Angle ICC_(C,1)_ (95% CI)
*Basketball*	ICC_(C,1)_ (95% CI)	0.963 [0.945, 0.981]	0.999 [0.998, 1.000]	0.981 [0.972, 0.989]	0.335 [0.187, 0.483]
	RMSE (95% CI)	0.007 [0.006, 0.009]	0.005 [0.004, 0.006]	0.023 [0.021, 0.025]	5.444 [3.60, 7.28]
*Mole Patrolll*	ICC_(C,1)_ (95% CI)	0.995 [0.994, 0.996]	0.998 [0.998, 0.999]	0.780 [0.662, 0.897]	0.986 [0.979, 0.993]
	RMSE (95% CI)	0.063 [0.053, 0.072]	0.069 [0.061, 0.077]	0.016 [0.012, 0.020]	20.3 [16.7, 23.9]
*Puzzle Walk*	ICC_(C,1)_ (95% CI)	0.993 [0.990, 0.995]	0.996 [0.995, 0.997]	0.976 [0.970, 0.983]	0.957 [0.924, 0.990]
	RMSE (95% CI)	0.050 [0.042, 0.058]	0.056 [0.050, 0.061]	0.027 [0.023, 0.030]	21.6 [19.1, 24.0]
*Smash*	ICC_(C,1)_ (95% CI)	0.932 [0.894, 0.971]	0.996 [0.995, 0.997]	0.736 [0.619, 0.853]	0.997 [0.993, 1.000]
	RMSE (95% CI)	0.032 [0.026, 0.037]	0.080 [0.076, 0.085]	0.011 [0.008, 0.013]	14.1 [11.7, 16.5]
*Wobbly Waiter*	ICC_(C,1)_ (95% CI)	0.984 [0.976, 0.992]	0.998 [0.998, 0.999]	0.998 [0.997, 0.998]	0.996 [0.995, 0.998]
	RMSE (95% CI)	0.041 [0.033, 0.050]	0.056 [0.053, 0.060]	0.013 [0.011, 0.015]	12.0 [10.3, 13.7]

**Table 4 sensors-25-07172-t004:** Motor-state segment classification statistics.

Motor State	Theia3D Segments	AR Segments	True Positives	False Positives	False Negatives	Precision	Recall	F1-Score	Mean Overlap ± SD
Straight Walking	130	134	125	5	9	0.962	0.933	0.947	97.1% ± 0.3%
Turning	114	114	113	1	1	0.991	0.991	0.991	78.3% ± 0.9%
Squatting	134	134	134	0	0	1.000	1.000	1.000	97.2% ± 0.6%
Sit-to-stand	39	39	39	0	0	1.000	1.000	1.000	90.8% ± 0.4%
Stand-to-sit	40	40	40	0	0	1.000	1.000	1.000	93.1% ± 0.8%

**Table 5 sensors-25-07172-t005:** Concurrent validity statistics for absolute agreement in mobility metrics between systems.

Motor State and Mobility Metric	Mean ± SD (SEM)	Mean ± SD (SEM)	Bias (95% Limits of Agreement)	*t*-Statistics or Wilcoxon Signed-Rank-Statistics ^1^	ICC_(A,1)_
**Straight walking**	**AR**	**Theia3D head**			
Step length (m)	0.61 ± 0.11 (0.03)	0.60 ± 0.11 (0.03)	−0.01 (−0.09 0.08)	t(14) = 0.71, *p* = 0.489	0.938
Max. gait speed (m/s)	1.66 ± 0.35 (0.01)	1.66 ± 0.35 (0.01)	0.00 (−0.03 0.03)	t(14) = −0.62, *p* = 0.543	0.999
Cadence (steps/min)	118.59 ± 14.04 (5.44)	120.37 ± 14.58 (5.65)	1.79 (−13.70 17.27)	t(14) = −0.88, *p* = 0.396	0.850
**Turning**	**AR**	**Theia3D pelvis**			
Turn duration (s)	2.05 ± 0.22 (0.13)	2.07 ± 0.20 (0.09)	0.02 (−0.17 0.20)	t(14) = −0.69, *p* = 0.504	0.904
Peak angular velocity (deg/s)	136.98 ± 16.95 (4.24)	116.06 ± 18.24 (4.54)	−20.92 (−42.34 0.51)	t(14) = 7.41, ***p* < 0.001**	0.477
**Squatting**	**AR**	**Theia3D head**			
Squat duration (s)	2.12 ± 0.57 (0.04)	2.11 ± 0.58 (0.04)	−0.01 (−0.12 0.10)	t(14) = 0.82, *p* = 0.425	0.995
Squat depth (m)	0.49 ± 0.18 (0.03)	0.45 ± 0.16 (0.03)	−0.03 (−0.08 0.02)	t(14) = 4.75, ***p* < 0.001**	0.972
**Transfers**	**AR**	**Theia3D head**			
Sit-to-stand duration (s)	1.20 ± 0.20 (0.03)	1.21 ± 0.17 (0.03)	0.00 (−0.07 0.08)	t(14) = −0.44, *p* = 0.664	0.979
Stand-to-sit duration (s)	1.57 ± 0.38 (0.06)	1.59 ± 0.41 (0.06)	0.01 (−0.15 0.18)	t(14) = −0.67, *p* = 0.513	0.978

^1^ Significant *p*-values in bold.

## Data Availability

The datasets and pseudocode supporting the findings of this study are publicly available in the GitHub repository https://github.com/pvand/Motorstate_detection_and_metrics and can be cited using the DOI: https://doi.org/10.5281/zenodo.17592651.

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
