# Peer review of "Deriving Motor States and Mobility Metrics from Gamified Augmented Reality Rehabilitation Exercises in People with Parkinson’s Disease"

_sensors, 2025, doi:10.3390/s25237172_

Round 1
Reviewer 1 Report
Comments and Suggestions for Authors
This manuscript presents a promising technological application. However, in its current form, it reads more like a technical validation report from the company than a dispassionate, critical scientific paper. To be accepted in a reputable journal, you must address these major conceptual, methodological, and interpretive flaws. The core idea is excellent, but the execution requires significant rigor and humility to meet the standards of the field. Focus on what your data actually shows, not what you hope it implies.
-
This is a fundamental conceptual flaw. You are comparing the AR system against Theia3D, but Theia3D is itself an algorithmic pipeline (with its own filters, inverse kinematics, and model fitting). It is not a direct measurement system like a marker-based system or force plates. You must explicitly frame this as a comparison between two algorithmic systems, not a validation against a "reference." The discussion must be tempered to acknowledge that any discrepancies could be due to errors in either system, not just the AR glasses.
-
A convenience sample of 15 participants, all in early disease stages (H&Y 1-2), is insufficient to make broad claims about "people with Parkinson's disease." The results are not generalizable to the wider PD population, especially those with more severe gait impairment, postural instability, or dyskinesia. This severely limits the clinical relevance and impact of your findings. You must explicitly state this as a major limitation and avoid over-generalizing your conclusions.
-
The pitch correction model (L = 0.2) is described as "empirically found." This is not scientifically rigorous. It appears to be a post-hoc correction to force the AR data to align with Theia3D, rather than a biomechanically validated model. This ad-hoc approach undermines the validity of all vertical metrics (step length, squat depth) and introduces an unquantified source of error. You need to provide a proper derivation or validation of this correction factor, or remove it and discuss the inherent limitations of the head-mounted origin.
-
Your classification relies heavily on rigid, empirically optimized thresholds (e.g., for turning and straight walking). This approach is brittle and unlikely to generalize to different environments, individuals, or game variations. A more robust approach would use a machine learning model trained on a larger, more diverse dataset. The high accuracy you report is likely a product of this over-fitting to your specific, controlled lab setup and a homogeneous patient group.
-
You state you used Theia3D pelvis yaw as the reference to ensure the AR system could distinguish full-body turns, but then you proceed to use AR head yaw for your final turn metrics. Your justification—that it's a "practical alternative"—is weak. The significant bias in peak angular velocity (~18%) directly proves that head yaw is a poor proxy for whole-body turning velocity due to extraneous movements. Your own data contradicts your methodology choice.
-
While you report ICCs, the wide Limits of Agreement (LoA) for metrics like cadence are clinically more important and tell a different story. Excellent ICC can hide poor absolute agreement, especially in homogeneous samples. Furthermore, you do not report the standard error of measurement (SEM) or the minimal detectable change (MDC), which are critical for interpreting whether the observed agreement is precise enough to track individual patient progress over time.
-
You argue that AR exercises can assess "real-world" mobility, yet the study was conducted in a single, controlled gait lab visit. This is no more ecologically valid than a standard clinical test. The claim that this approach can capture "continuous, objective data on motor performance during daily exercises" is not supported by your data, which only captures short, supervised gameplay snippets. You have not demonstrated performance in a true home environment.
-
You dismiss the only moderate agreement for peak turn velocity and the significant bias in squat depth as minor issues with straightforward explanations. This is not critical enough. These are major shortcomings that limit the utility of these specific metrics. The discussion should devote more space to the implications of these poorer agreements and what they mean for the limitations of a head-worn sensor.
-
The descriptions of the algorithms in sections 2.5.1-2.5.4 are dense and difficult to follow. A flowchart for each motor state detection algorithm would be vastly more effective. Furthermore, the thresholds in Table 2 are presented without a clear explanation of the optimization process or the cost function used in the "iterative error-cost approach." The methodology is not easily reproducible.
-
With two authors (including the senior author) having financial stakes in Stroll Ltd., the company that produces the AR platform being tested, there is a high risk of perceived (or actual) bias. The manuscript's overly optimistic tone in the abstract and conclusions exacerbates this. The language must be made more neutral and conservative. The limitations section should explicitly address how the COI was managed to ensure analytical objectivity (e.g., were the algorithms and statistical thresholds blinded or pre-registered?).
Reviewer 2 Report
Comments and Suggestions for Authors
This paper analyses and compares mobility metrics from augmented reality (AR) glasses against a markerless motion capture system in people with Parkinson’s disease. AR-based assessments showed high agreement and accuracy in identifying motor states. The main results support AR glasses as a tool for monitoring mobility during clinical or home rehabilitation.
The paper presents several analyses comparing two strategies for monitoring PD patients. The analyses are interesting, but I think the technical contribution is low. It seems more like a paper describing a new dataset than a technical one.
- I’d suggest including a list of contributions at the end of the introduction.
- Regarding the dataset, Do you have a ground truth? There are labels generated by experts.
- It seems that you only compare strategies for obtaining information for state classification. I think it would be necessary to have a ground truth to see the errors generated by each strategy
- The ergonomics of using glasses compared to using Theia3D can be analysed. Using glasses can be more invasive while the use of cameras can be more intrusive?
- How are the differences between subjects compared to the differences between systems?
- Review figures' references in the text
Well written
Round 2
Reviewer 1 Report
Comments and Suggestions for Authors
accept
Author Response
We thank the reviewer for their time and for accepting the revised manuscript. We appreciate your feedback and are glad that our revisions meet your expectations.
Reviewer 2 Report
Comments and Suggestions for Authors
I appreciate the important effort for improving the paper but I have a minor aspect. Regarding the comments not addressed like those related to the ground truth, I’d suggest including them in a limitation subsection.
Author Response
We thank the reviewer for their comment. As noted in our initial response, the motor states in this dataset were derived from our own analysis pipeline applied to synchronized AR and Theia3D recordings, and no independent expert labels were available. This limitation is discussed in the manuscript’s limitations section:
“Second, the study lacked a definitive ground truth for motor-state detection. Without labeled reference data, the validation of AR-derived outcomes relies on indirect comparisons rather than true state-by-state agreement. Establishing ground-truth labels through for example expert video annotation and applying super- vised machine-learning methods could potentially provide a more robust basis for validating AR-derived motor states. Unfortunately, the current dataset was not large enough to support such supervised machine-learning approaches.”
Therefore, we believe that this limitation has already been appropriately addressed in the manuscript, and no additional revisions are necessary regarding this point.